# EVALUATING ROBUSTNESS OF COOPERATIVE MARL

## ABSTRACT

In recent years, a proliferation of methods were developed for multi-agent reinforcement learning (MARL). In this paper, we focus on evaluating the robustness of MARL agents in continuous control tasks. In particular, we propose the first model-based approach to perform adversarial attacks for continuous MARL. The attack aims at adversarially perturbing the states of agent(s) to misguide them to take damaging actions that lower team rewards. A deep neural network is trained to represent the dynamics of the environment. We then solve an optimization problem with the learned dynamics model to yield small perturbations. In addition, we discuss several strategies to optimally select adversary agents for the attack. Numerical experiments on multi-agent Mujoco tasks verify the effectiveness of our proposed approach.

## 1 INTRODUCTION

Deep neural networks are known to be vulnerable to adversarial examples, where a small and often imperceptible adversarial perturbation can easily fool the state-of-the-art deep neural network classifiers (Szegedy et al., 2013; Nguyen et al., 2015; Goodfellow et al., 2014; Papernot et al., 2016). Since then, a wide variety of deep learning tasks have been shown to also be vulnerable to adversarial attacks, ranging from various computer vision tasks to natural language processing tasks (Jia & Liang, 2017; Zhang et al., 2020b; Jin et al., 2020; Alzantot et al., 2018).

Perhaps unsurprisingly, deep reinforcement learning (DRL) agents are also vulnerable to adversarial attacks, as first shown in Huang et al. (2017) for atari games DRL agents. In Huang et al. (2017), the authors study the effectiveness of adversarial examples on a policy network trained on Atari games under the situation where the attacker has access to the neural network of the target policy. In Lin et al. (2017), the authors further investigate a strategically-time attack by attacking trained agents on Atari games at a subset of the time step. Meanwhile, Kos & Song (2017) use the fast gradient sign method (FGSM) to generate adversarial perturbation on the A3C algorithm (Mnih et al., 2016) and explore training with random noise and FGSM perturbation to improve resilience against adversarial examples. While the above research endeavors focused on actions that take discrete values, another line of research tackles a more challenging problem on DRL with continuous action spaces (Weng et al., 2019; Gleave et al., 2019). Specifically, Weng et al. (2019) consider a two-step algorithm which determines adversarial perturbation to be closer to a target state using a learnt dynamics model, and Gleave et al. (2019) propose a physically realistic thread model and demonstrates the existence of adversarial policies in zero-sum simulated robotics games.

While most of the existing DRL attack algorithms focus on the *single* DRL agent setting[1], in this work we propose to study the vulnerability of *multi-agent* DRL, which has been widely applied in many safety-critical real-world applications including swarm robotics (Dudek et al., 1993), electricity distribution, and traffic control (OroojlooyJadid & Hajinezhad, 2019). In particular, we focus on the collaborative multi-agent reinforcement learning (c-MARL) setting, where a group of agents is trained to generate joint actions to maximize certain (team) reward function. We note that c-MARL is a more challenging yet interesting setting than the *single* DRL agent setting, as the interactions between agents commands consideration of additional layers of complications.

**Contribution.** Our contribution can be summarized as follows:

---

[1]We are aware of only one recent work (Lin et al., 2020) on attacking c-MARL agents with *discrete* action space, rather than *continuous* action space considered in our work.

- We propose a new adversarial attack framework on MARL with continuous action space, where we name it cMBA (model-based attack on c-MARL). The attack comprises two steps: learning a representation of the environment dynamics and then determining proper observation perturbations based on the learned dynamics model.

- We formulate the process of selecting an victim agent as a mixed-integer programming problem and propose an approximate formulation that can be efficiently solved by a first-order method.

- We evaluate our attack framework with two baselines and four multi-agent MuJoCo tasks (Peng et al., 2020), with number of agents ranging from 2 to 6, to verify the effectiveness of our approach. Our model-based attack consistently outperforms the two baselines in all tested environments.

**Paper outline.** Section 2 discusses related works in adversarial attacks for MARL along with the problem settings. We describe the main attack framework cMBA in Section 3, including the training of the dynamics model and solving the sub-optimization problem. In Section 3.3, we introduce a mixed-integer program to find the optimal set of victim agents and propose an approximate formulation that can be efficiently solved using a first-order optimization method. Section 4 presents the evaluation of our approach on several multi-agent MuJoCo environments. Therein, we also study the performance of different variants of our model-based attack as discussed in Section 3.3.

## 2   RELATED WORK

**Adversarial attacks for DRL agents.**   In the c-MARL setting, although there exists a significant literature focusing on the adversarial training of MARL agents (Phan et al., 2020; 2021; Zhang et al., 2020a), yet a very few of them addresses the scenario of adversarial attacks in c-MARL. Lin et al. (2020) propose a two-step attack on a single agent in a c-MARL environment, where they extend existing methods to generate adversarial examples guided by a learned adversarial policy. The attack in Lin et al. (2020) is evaluated under the StarCraft Multi-Agent Challenge (SMAC) environment (Samvelyan et al., 2019) where the action spaces are discrete. To the best of our knowledge, there has not been work considering adversarial attacks on the c-MARL setting with continuous action spaces. The closest approach to ours is the enchanting attack in Lin et al. (2017) and the model-based attack in Weng et al. (2019); however, these two works focused on *single* DRL agent setting. The enchanting attack in Lin et al. (2017) uses a video prediction model to predict the future state and finds perturbation that minimizes a distance function between the predicted future state and a target state. Similarly, Weng et al. (2019) consider two formulations for observation and action perturbation that incorporates a learned dynamic model for future state prediction. In addition to the model-based approach, in this paper, we consider the selection of victim agents, which is unique to the multi-agent DRL setting.

**Training c-MARL agents.**   One simple approach to train c-MARL agents is to let each agent learn their own policy independently as in Independent Q-Learning (IQL) (Tan, 1993; Tampuu et al., 2017). However, this training strategy does not capture the interaction between agents. Alternatively, one can follow a common paradigm called centralized training with decentralized execution (CTDE) (Oliehoek et al., 2008; Oliehoek & Amato, 2016; Kraemer & Banerjee, 2016). Gupta et al. (2017) extend DQN (Mnih et al., 2015), TRPO (Schulman et al., 2015), and DDPG (Lillicrap et al., 2015) to c-MARL, where a parameter sharing neural network policy is trained in a centralized fashion but still allows decentralized behavior across agents. Similar to this, COMA (Foerster et al., 2018) is an actor-critic method that uses a centralized critic to estimate counterfactual advantage for decentralized policies. While COMA uses a single critic for all agents, MADDPG (Lowe et al., 2017) learns a centralized critic for each agent and evaluates on continuous control. Instead of having fully centralized critic, DOP (Wang et al., 2020b) and FACMAC (Peng et al., 2020) apply the value decomposition idea (Sunehag et al., 2017; Rashid et al., 2018; 2020; Son et al., 2019; Wang et al., 2020a) into multi-agent actor-critic by having a centralized but factorized critic.

## 3 CMBA: MODEL-BASED ATTACK FOR C-MARL

### 3.1 PROBLEM SETTING

We consider multi-agent tasks with continuous action spaces modeled as a Decentralized Partially Observable Markov Decision Process (Dec-POMDP) (Oliehoek & Amato, 2016). A Dec-POMDP has a finite set of agents $\mathcal{N} \equiv \{1, \cdots, n\}$ associated with a set of states $\mathcal{S}$ describing global states, a set of continuous actions $\mathcal{A}_i$, and a set of individual partial observation $\mathcal{O}_i$ for each agent $i \in \mathcal{N}$. Given the current observation $o_t^i \in \mathcal{O}_i$, the action $a_t^i \in \mathcal{A}_i$ is selected by a parametrized policy $\pi_{\theta_i} : \mathcal{O}_i \rightarrow \mathcal{A}_i$. The next state agent $i$ moves to is determined by the state transition function $\mathcal{P}_i : \mathcal{S} \times \mathcal{A}_i \rightarrow \mathcal{S}$. After that, agent $i$ receives a reward $r_t^i$ calculated from a reward function $\mathcal{R}_i : \mathcal{S} \times \mathcal{A}_i \rightarrow \mathbb{R}$ and observes a local observation $o_{t+1}^i \in \mathcal{O}_i$ correlated with the global state $s^{t+1} \in \mathcal{S}$. In addition, Dec-POMDP is associated with an initial state distribution $\mathcal{P}_0$ and a discount factor $\gamma$. In this setting, each agent tries to find the policy that maximizes its own total expected return $R_i = \sum_{t=0}^T \gamma^t r_t^i$ where $T$ is the sampling horizon.

### 3.2 PROBLEM FORMULATION

Our goal is to generate adversarial perturbations imposed to the victim agents' input (state) in order to deteriorate the total team reward. The added noise encourages the victim agents' observation to be close to a desired adversarial state corresponding to low reward. To avoid sampling from the environment, we use a pre-trained model that learns the dynamics of the environment to predict the next state from the perturbed observation and current action, then find the suitable noise that minimizes the distance between the predicted next state and a predefined target state. This process can be formulated as an optimization problem as follows.

Formally, we consider a multi-agent setting with $|\mathcal{N}| = n$ agents, each agent $i \in \mathcal{N}$ observes state $s_t^i$ locally and takes action $a_t^i$. Let $s_t = (s_t^1, \cdots, s_t^n) \in \mathcal{S}$ be the global state at time step $t$ which is concatenated from local states $s_t^i$ for each agent $i = 1, \cdots, n$. We also denote the joint action $a_t = (a_t^1, \cdots, a_t^n)$ concatenated from each agent's action $a_t^i$. Let $\mathcal{L}_t \in \mathcal{N}$ be the set of victim agents at time step $t$, i.e. the set of agents that can be attacked. Let $f_d : \mathcal{S} \times \mathcal{A} \rightarrow \mathcal{S}$ be a parametrized function that approximates the dynamics of the environment, where $\mathcal{A}$ is the set of concatenated actions, one from each $\mathcal{A}_i$. We assume that a pre-trained MARL policy $\pi(\cdot)$ is given. Let $s_{target}$ be the target state which can lead to poor performance to the agent. We denote $\epsilon$ as an upper bound on budget constraint w.r.t some $p$-norm $\|\cdot\|_p$. The state perturbation $\Delta s = (\Delta s^1, \cdots, \Delta s^n)$ (we suppress the dependence on $t$ of $\Delta s$ to avoid overloading the notation) to $s_t$ is the solution to the following problem

$$
\begin{aligned}
\min_{\Delta s=(\Delta s^1, \cdots, \Delta s^n)} \quad & d(\hat{s}_{t+1}, s_{target}) \\
\text{s.t.} \quad & \hat{s}_{t+1} = f_d(s_t, a_t) \\
& a_t^i = \pi(s_t^i + \Delta s^i) \\
& \Delta s^i = \mathbf{0}, \ \forall \, i \notin \mathcal{L}_t \\
& \ell_{\mathcal{S}} \leq s_t + \Delta s \leq u_{\mathcal{S}} \\
& \|\Delta s\|_p \leq \epsilon
\end{aligned}
\tag{1}
$$

where $\mathbf{0}$ is a zero vector, and the observation space is constrained within the (vectorized) intervals between $\ell_{\mathcal{S}}$ and $u_{\mathcal{S}}$.

Let us first provide some insights for the formulation (1). For each agent, using the trained policy $\pi$, we can compute the corresponding action $a_t^i$ given its (possibly perturbed) local state $s_t^i$ or $s_t^i + \Delta s^i$. From the concatenated state-action pair $(s_t, a_t)$, we can predict the next state $\hat{s}_{t+1}$ via the learned dynamics model $f_d$. Then by minimizing the distance between the predicted next state and the target state under the budget constraint, we are forcing the agent to move closer to a damaging state in the next transition. We also note that the last equality constraint indicates that we only allow perturbing agents within the victim set $\mathcal{L}_t$.

Problem (1) can be efficiently solved by proximal-gradient-based methods. Firstly, by substituting the definition of $a_t$ into $\hat{s}_{t+1}$, (1) is equivalent to

$$
\begin{aligned}
\min_{x} \quad & d(f_d(s_t, \pi(s_t + x)), s_{target}) \\
\text{s.t.} \quad & x \in \mathcal{C}_t^p
\end{aligned}
\tag{2}
$$

where $\mathcal{C}_t^p := \{x = (x_1, \cdots, x_n) : \|x\|_p \leq \epsilon, \ \ell_{\mathcal{S}} - s_t \leq x \leq u_{\mathcal{S}} - s_t \text{ and } x_i = \mathbf{0} \text{ for } i \notin \mathcal{L}_t\}$. If we choose the distance function as $d(a, b) = \|a - b\|^2$, we can use the projected gradient descent (PGD) algorithm (Nesterov, 2003) to solve (2). The PGD iteration to update $y_k$ at iteration $k$ can be described as

$$y_{k+1} = \text{proj}_{\mathcal{C}_t^p} [y_k - \eta \nabla_y d(f_d(s_t, \pi(s_t + y)), s_{target})] \tag{3}$$

where $\text{proj}_{\mathcal{C}_t^p}(\cdot)$ is the projection to the convex set $\mathcal{C}_t^p$ and $\eta$ is the learning rate. The projection is simple to calculate since $\mathcal{C}_t^p$ is the intersection of a unit ball in the $p$-norm and a box.

The whole attack process can be summarized in Algorithm 1.

---

**Algorithm 1** cMBA algorithm at timestep $t$

---

1: **Initialization:**
    Given $s_t, s_{target}, \pi, f_d, \mathcal{L}_t$; initialize $\Delta s_0$; choose learning rate $\eta > 0$
2: **For** $k := 0, \cdots, K - 1$ **do**
3:    Compute $a_t = (a_t^1, \cdots, a_t^n)$ as

$$a_t^i = \begin{cases} \pi(s_t^i + \Delta s^i) & \text{if } i \in \mathcal{L}_t \\ \pi(s_t^i) & \text{otherwise.} \end{cases}$$

4:    Compute $\hat{s}_{t+1} = f_d(s_t, a_t)$
5:    Update $\Delta s$ as

$$\Delta s_{k+1} = \text{proj}_{\mathcal{C}_t} [\Delta s_k - \eta \nabla_{\Delta s} d(\hat{s}_{t+1}, s_{target})]$$

6: **End For**

---

**Learning dynamics model.** The most important factor that affects the solution of (2) is the quality of the learned dynamics model $f_d$. If the dynamics is known, we can solve (2) exactly so that the solution is indeed the optimal perturbation. However, it is hardly the case in practice because we often approximate the dynamics using function approximators such as neural networks. The parameter $w$ for $f_d$ is the solution of the following optimization problem

$$\min_w \ \sum_{i \in \mathcal{D}} \left\| f_d(s_{cur}^i, w) - s_{next}^i \right\|^2 \tag{4}$$

where $\mathcal{D}$ is a collection of state transition $\{(s_{cur}^i, s_{next}^i)\}_{i=1}^{|\mathcal{D}|}$ where $s_{next}$ is the actual state that the environment transitions to after taking action $a_t$ determined by a given policy. In particular, we separately collect transitions using the pre-trained policy $\pi$ and a random policy to obtain $\mathcal{D}_{train}$ and $\mathcal{D}_{random}$. Then the dataset $\mathcal{D}$ is built as $\mathcal{D} = \mathcal{D}_{train} \cup \mathcal{D}_{random}$. Problem (4) is a standard supervised learning problem and can be solved using gradient-based learning. We describe the full process of training the dynamics model in Algorithm 2 where the `GradientBasedUpdate` step is any gradient-descent-type update.

---

**Algorithm 2** Training dynamics model

---

1: **Initialization:**
    Given pre-trained policy $\pi_{tr}$ and a random policy $\pi_{rd}$; initialize $w_0$
2: Form $\mathcal{D} := \mathcal{D}_{train} \cup \mathcal{D}_{random}$ by collecting a set of transitions $\mathcal{D}_{train}$ and $\mathcal{D}_{random}$ using policy $\pi_{tr}$ and $\pi_{rd}$, respectively.
3: **For** $k := 0, 1, \cdots$ **do**

$$w_{k+1} = \texttt{GradientBasedUpdate}(\mathcal{D}, w_k)$$

4: **End For**

---

### 3.3 VICTIM AGENT SELECTION

In this subsection, we discuss different strategies to select victim agents when performing adversarial attacks in the MARL setting. Suppose that we would like to perform an attack on any agent but the budget on the victim agents is $n_a \leq n$. In other words, an attacker can only choose a subset of

agents to attack during each time step, and the goal of an attacker is to choose the most vulnerable agents so that the total reward is utmost affected. We note that this scenario is unique in the setting of *multi-agent* DRL setting, as in the *single* DRL agent setting we can only attack the same agent all the time. To start with, we first formulate a mixed-integer program to perform the attack on a set of optimally victim agents as below:

$$
\begin{aligned}
\min_{\Delta s, w} \quad & d(\hat{s}_{t+1}, s_{target}) \\
\text{s.t.} \quad & \hat{s}_{t+1} = f_d(s_t, a_t) \\
& a_t = \pi(s_t + \langle \Delta s, w \rangle) \\
& \ell_{\mathcal{S}} \leq s_t + \Delta s \leq u_{\mathcal{S}} \\
& \|\Delta s\|_p \leq \epsilon \\
& w_i \in \{0, 1\} \\
& \sum_i w_i = n_a
\end{aligned}
\tag{5}
$$

where we introduce a new binary variable $w$ to select the appropriate agents' input to perturb and $n_a$ is the total number of victim agents we can attack.

Due to the existence of the new binary variable, problem (5) is much harder to solve than before. We instead solve a proxy of (5) as follows

$$
\begin{aligned}
\min_{\Delta s, \theta} \quad & d(\hat{s}_{t+1}, s_{target}) \\
\text{s.t.} \quad & \hat{s}_{t+1} = f_d(s_t, a_t) \\
& a_t = \pi(s_t + \langle \Delta s, W(s_t, \theta) \rangle) \\
& \ell_{\mathcal{S}} \leq s_t + \Delta s \leq u_{\mathcal{S}} \\
& \|\Delta s\|_p \leq \epsilon \\
& 0 \leq W(s_t, \theta) \leq 1
\end{aligned}
\tag{6}
$$

where $W(\cdot, \theta)$ is a function parametrized by $\theta$ that takes current state as input and outputs the weight to distribute the noise to each agent. Suppose we represent $W(\cdot, \theta)$ by a neural network, we can rewrite the formulation (6) as

$$
\begin{aligned}
\min_{\Delta s, \theta} \quad & d(f_d(s_t, \pi(s_t + \langle \Delta s, W(s_t, \theta) \rangle)), s_{target}) \\
\text{s.t.} \quad & \Delta s \in \mathcal{C}_t^p
\end{aligned}
\tag{7}
$$

because the last constraint in (6) can be enforced by using a softmax activation in the neural network $W(\cdot, \theta)$. As a result, (7) can be efficiently solved by using PGD. We present the pseudo-code of the attack in Algorithm 3. After $K$ steps of PGD update, we define the index $i_{n-j}$ as the $j$-th largest value within $W(s_t, \theta_K) \in \mathbb{R}^n$, i.e. we have $W_{i_{(n)}}(s_t, \theta_K) \geq W_{i_{(n-1)}}(s_t, \theta_K) \geq \cdots \geq W_{i_{(1)}}(s_t, \theta_K)$. Let $\mathcal{I}_j$ be the index set of top-$j$ largest outputs of the $W(s_t, \theta_K)$ network. The final perturbation returned by our victim agent selection strategy will be $\widehat{\Delta s} = ((\widehat{\Delta s})^1, \cdots, (\widehat{\Delta s})^n)$ where $(\widehat{\Delta s})^i = \mathbf{0}$ if $i \notin \mathcal{I}_{n_a}$ and $(\widehat{\Delta s})^i = (\Delta s_K)^i$ if $i \in \mathcal{I}_{n_a}$.

---

**Algorithm 3** cMBA with victim agent selection at timestep $t$

---

1: **Initialization:**
    Given $s_t, s_{target}, \pi, f_d, n_a$; initialize $\Delta s_0$; choose learning rate $\eta > 0 \; \lambda > 0$.
2: **For** $k := 0, \cdots, K - 1$ **do**
3:      Compute $a_t = \pi(s_t + \langle \Delta s, W(s_t, \theta) \rangle)$
4:      Compute $\hat{s}_{t+1} = f_d(s_t, a_t)$
5:      Update $\Delta s$: $\Delta s_{k+1} = \text{proj}_{\mathcal{C}_t^p} [\Delta s_k - \eta \nabla_{\Delta s} d(\hat{s}_{t+1}, s_{target})]$
6:      Update $\theta$: $\theta_{k+1} = \theta_k - \lambda \nabla_\theta d(\hat{s}_{t+1}, s_{target})$
7: **End For**
8: Compute $\mathcal{I}_{n_a} = \{i_{(n)}, \cdots, i_{(n-n_a)}\}$ such that

$$
W_{i_{(n)}}(s_t, \theta_K) \geq W_{i_{(n-1)}}(s_t, \theta_K) \geq \cdots \geq W_{i_{(1)}}(s_t, \theta_K)
$$

9: Return $\widehat{\Delta s} = ((\widehat{\Delta s})^1, \cdots, (\widehat{\Delta s})^n)$ such that

$$
(\widehat{\Delta s})^i = \begin{cases} (\Delta s_K)^i & \text{if } i \in \mathcal{I}_{n_a} \\ \mathbf{0} & \text{otherwise} \end{cases}
$$

---

**Remark 3.1** *For this attack, we assume each agent $i$ has access to the other agent's observation to form the concatenated state $s_t$.*

## 4 EXPERIMENTS

We perform the attack on various multi-agent MuJoCo (MA-MuJoCo) environments including Walker2d (2x3), HalfCheetah (2x3), HalfCheetah (6x1), and Ant (4x2). The pair `environment name (config)` indicates the name of MuJoCo environment along with the agent partition, where a configuration of 2x3 means there are in total 2 agents and each agent has 3 actions. Note that we cannot directly apply methods such as the fast gradient sign method (FGSM) or JSMA (an attack using saliency map) as in Lin et al. (2020), since those methods require a "target" action which is not available in continuous action space. Therefore, we consider two baselines: `Uniform` and `Gaussian` baselines where the perturbation follows either Uniform distribution $U(-\epsilon, \epsilon)$ or Normal distribution $\mathcal{N}(0, \epsilon)$.

**Variants of model-based attack.** We consider the following variants of our model-based attack:

- Model-based attack on fixed agents: perform Algorithm 1 using a fixed set of victim agents $\mathcal{L}$.

- Best model-based attack on fixed agents: among the model-based attack on fixed agents, pick the subset that achieves the best performance.

- Model-based attack on random agents: perform Algorithm 1 with $\mathcal{L}_t$, which is sampled uniformly from $\mathcal{N}$ such that $|\mathcal{L}_t| = n_a$ ($n_a$ is the total victim agents).

- Model-based attack with learned victim selection: perform Algorithm 3 to select vulnerable agents and perform the attack on them.

- Greedy victim selection: perform a sweep over subsets of agents with size $n_a$. For each subset, perform Algorithm 1 to obtain proper perturbation and retrieve the corresponding objective value (distance between predicted state and target state). Select the subset corresponding to the lowest objective value as the victim agents.

**Experiment setup.** We first use MADDPG (Lowe et al., 2017) to train MARL agents for the four MA-MuJoCo environments listed above. Using the trained agents, we collect datasets containing one million transitions to train the dynamics model for each environment. The dynamics model is a fully connected neural network with three hidden layers of 1000 neurons. We also use a fully-connected neural network for $W(s_t, \theta)$ in (6) with two hidden layers of 200 neurons. We use AdamW (Loshchilov & Hutter, 2017) as the optimizer and select the best learning rate from $\{1, 5\} \times \{10^{-5}, 10^{-4}, 10^{-3}\}$ (the best learning rate is the one achieving lowest prediction error on a test set of $80,000$ samples). For our model-based attack, we run PGD for $K = 30$ steps to solve (2) and (5). We perform each attack over 16 episodes then average the rewards. We also illustrate the standard deviation of rewards using the shaded area in the plots.

**Model-free baselines vs model-based attack.** In this experiment, we run the two baseline attacks along with our model-based attack on the four MA-MuJoCo environments, in which there is only one victim agent ($n_a = 1$). Figure 1 illustrates the performance when we perform these attacks on specific agents under multiple budget levels using $\ell_\infty$-norm. For a fixed agent, our model-based attack consistently outperforms the two baselines. In particular, our model-based attack yields much lower rewards under low budget constraints (small $\epsilon$) compared to the two baselines.

To better visualize the performance difference, 2 illustrates the environment with and without attacks captured at different time-steps. From Figure 2, our model-based attack is able to make the MuJoCo agent flip, which terminates the episode at the 409-th timestep. The episode length and total rewards for each variant are: No attack$(1000, 2644.60)$, Uniform$(1000, 1758.97)$, Gaussian$(891, 1399.51)$, **Ours$(409, 287.06)$**.

We also investigate how the state values change during these attacks. Figure 3 presents different recordings of state values under adversarial attacks compared to no attack. Consider state index $8$, which represents the horizontal velocity of the agent. For the `HalfCheetah` environment, as the goal is to make the agent move forward as fast as possible, we expect the reward to be proportional to this state value. From Figure 3, all three attacks have fairly sparse red fractions across timesteps, which result in a much lower reward compared to the no-attack setting. Among the three attacks, our model-based ones appear to have the most sparse red fractions leading to the lowest rewards. In addition, the model-based attack appears to show its advantage in environments with more agents.

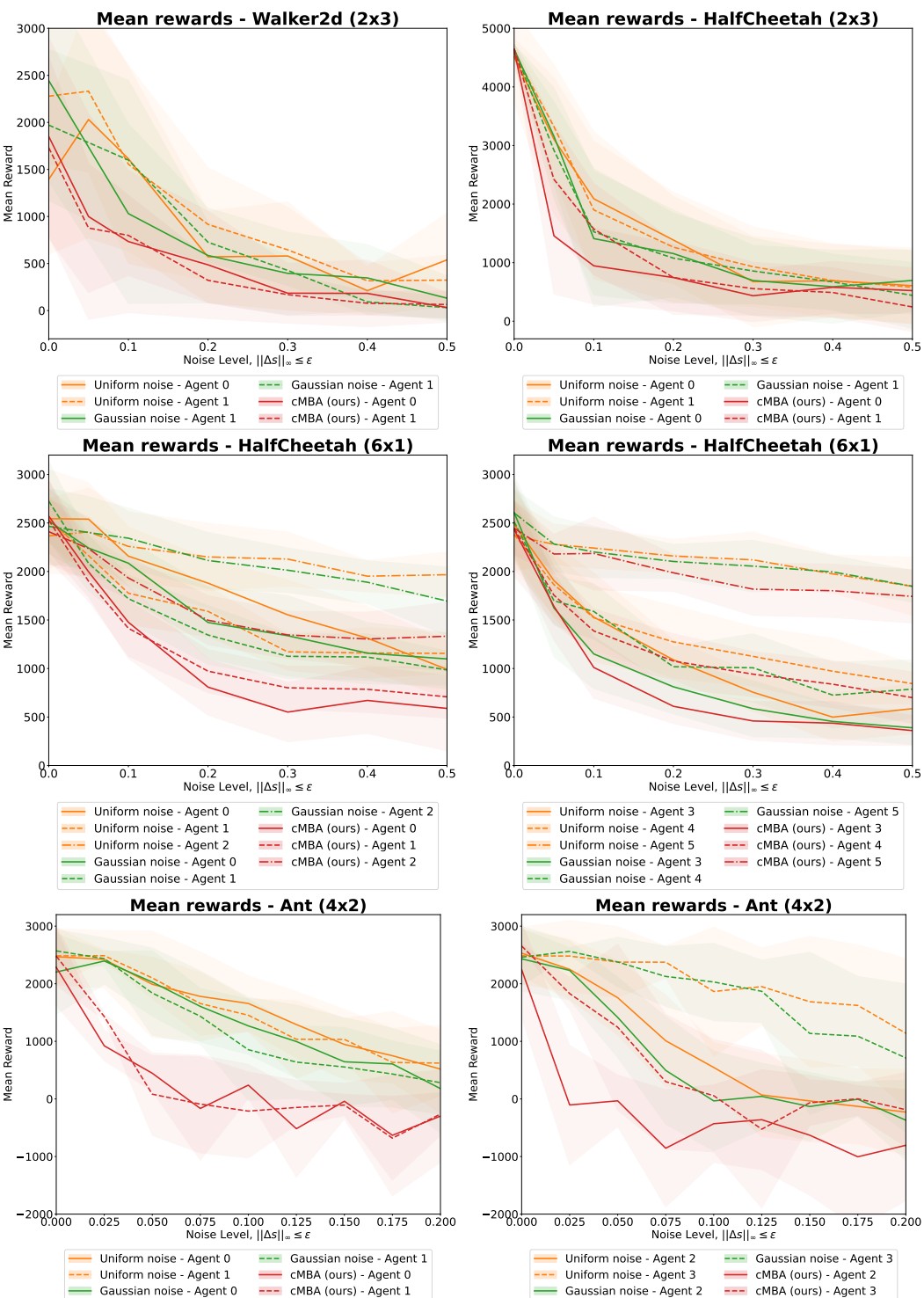

Figure 1: Model based attack vs baseline attacks on fixed agents.

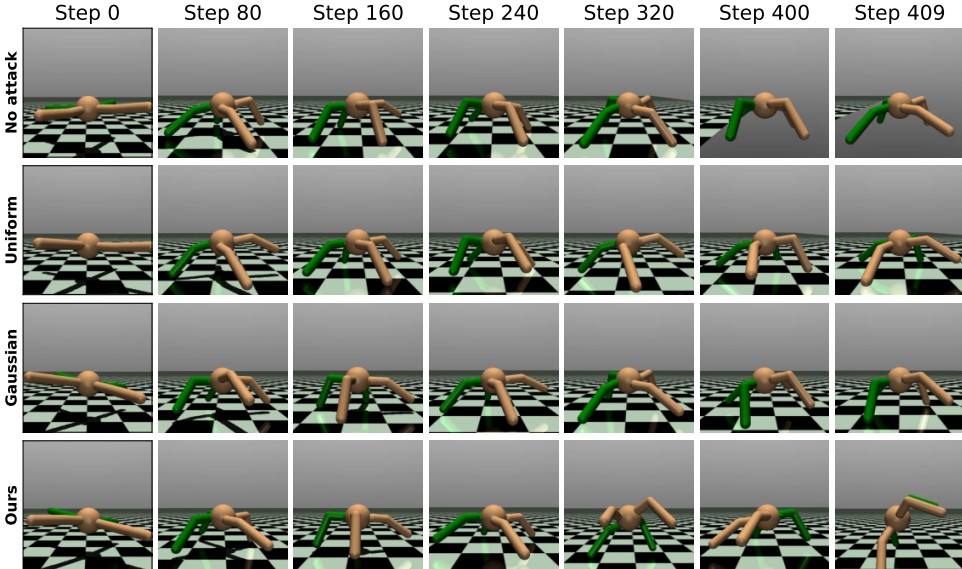

Figure 2: Various attacks on Agent $0$ in Ant (4x2) environment with $\|\Delta s\|_\infty \leq 0.1$. It can be seen that the agent flip at the end of episode under our model-based attack, demonstrating the effectiveness of our algorithm.

In particular, our approach results in lower rewards under a smaller budget as seen in HalfCheetah (6x1) and Ant (4x2) environments.

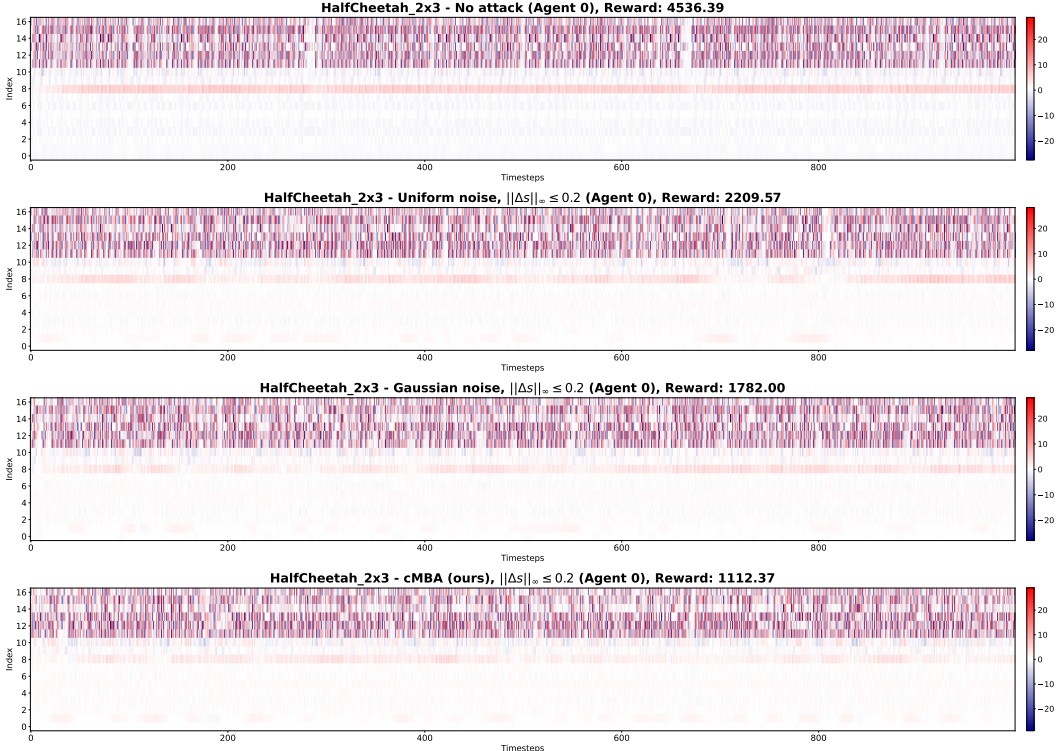

Figure 3: Recordings of state values in an episode under different attack on Agent $0$ in HalfCheetah( 2x3) environment.

**Effectiveness of learned victim selection.** We further demonstrate the performance of our model-based attack with and without learned victim selection on one victim agent. Figure 4 illustrates the performance of four model-based variants: the best performing model-based attack on the fixed agent (Best Fixed Agent), model-based attack on the random agent (Random Agent), model-based with learned victim selection as in Algorithm 3 (Learned victim Selection), and model-based with greedy victim selection (Greedy). Surprisingly, even though the greedy approach employs a brute-force strategy to select the agent with the closest distance, its performance reveals that selecting victim agents greedily might not be a good strategy. The learned victim selection seems to be better than the random or greedy strategy and is comparable with the random one in HalfCheetah environment. The best-fixed agent variant has good performance under low budget constraints in HalfCheetah (6x1) and Ant (4x2) environments. We want to emphasize that although the learned victim selection strategy in Section 3.3 is sub-optimal, it works well in these four environments.

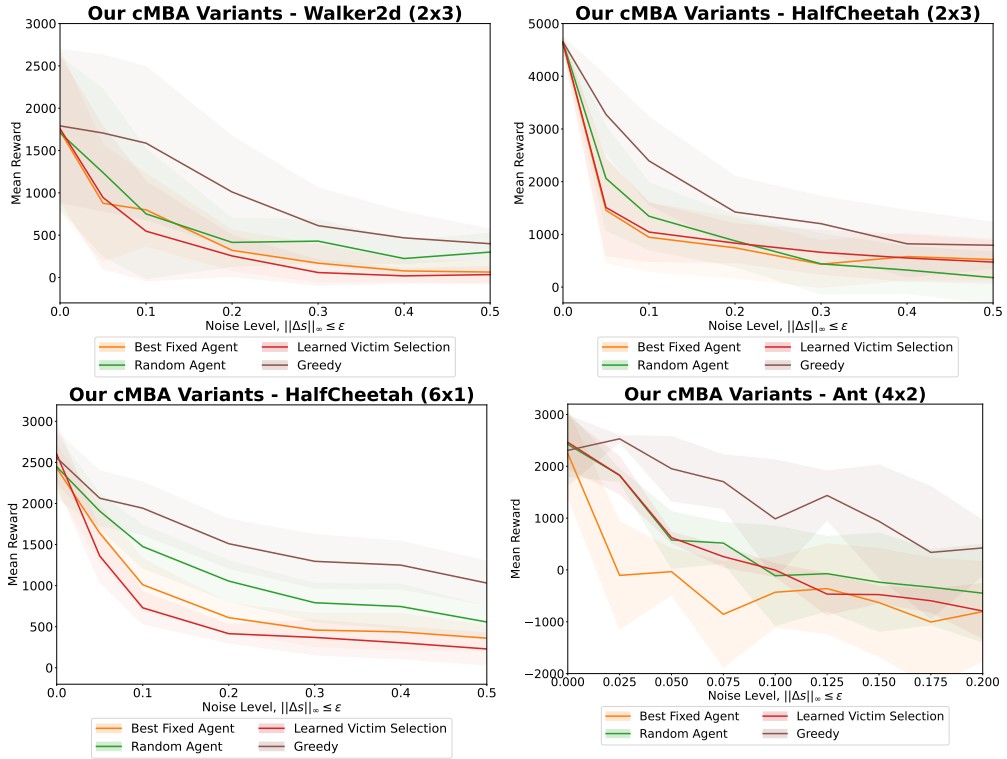

Figure 4: Performance of different variants of our proposed model-based attack (c-MBA) with different choices on victim agents.

## 5 CONCLUSIONS

In this paper, we propose a new attack algorithm named cMBA for evaluating the robustness of c-MARL environment with continuous action space. Our cMBA algorithm is the first to consider the c-MARL and the adversarial perturbation is computed by solving an optimization problem with learned dynamics models. The cMBA approach outperforms the other baselines attack by a large margin under 4 multi-agent MuJoCo environments especially ones with larger number of agents. Unique to multi-agent setting, we also study different strategies to select victim agents. Extensive experiment results on standard c-MARL benchmarks show that our proposed model-based attack and victim-selection strategy can successfully degrade the performance of well-trained c-MARL agents while outperforming other baselines by a large margin. A future direction is to consider a timed attack strategy where the perturbation is added at certain timesteps. We can also consider longer planning step, i.e. trying to reach a target state within the next $T$ steps instead of 1.

ETHICS STATEMENT

This paper does not contain ethics concerns.

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

## A    MODE DETAILS ON EXPERIMENT SETUP IN SECTION 4

**Specifying target state for each environment.** To perform our model based attack, we need to specify a target state that potentially worsens the total reward. In multi-agent MuJoCo environments, each agent has access to its own observation of the agent consisting the position-related and velocity-related information. The position-related information includes part of $x, y, z$ coordinates and the quarternion that represents the orientation of the agent. The velocity-related information contains global linear velocities and angular velocities for each joint in a MuJoCo agent. We refer the reader to Todorov et al. (2012) for more information about each MuJoCo environment. Now we describe the design of this target state for each environment as follows:

- **Walker (2x3)** environment: Since the episode ends whenever the agent falls, i.e. the $z$ coordinate falls below certain threshold. In this environment, the target state has a value of 0 for the index that corresponds to the $z$ coordinate of the MuJoCo agent (index 0).

- **HalfCheetah (2x3)** and **HalfCheetah (6x1)** environments: As the goal is to make agent moves as fast as possible, we set the index corresponding to the linear velocities to 0 (index 8).

- **Ant (4x2)** environment: As the agent can move freely in a 2D-plan, we set the index corresponding to the $x, y$ linear velocities to 0 (indices 13 and 14).

## B    ADDITIONAL EXPERIMENTS

In this section, we present experimental results in addition to ones presented in Section 4.

Figure 5 illustrates the environment with and without attacks captured at different timesteps. From Figure 5, our model based attack is able to make the MuJoCo agent fall down which terminates the episode at the 65-th timestep. The episode length and total rewards for each variant are: No attack$(478, 1736.44)$, Uniform$(382, 1037.73)$, Gaussian$(90, 32.24)$, **Ours$(63, -34.26)$**. Figure 6 illustrates how the state values change during an episode. These state values correspond to the agent shown in Figure 5. If we look at how the noise values change as in Figure 7, the noise generated by our approach appears to maximize the permissible noise budget.

In addition to the $\ell_\infty$-norm budget constraint, we also evaluate adversarial attacks using the $\ell_1$-norm constraint. Note that using $\ell_1$-norm for budget constraint is more challenging as the attack needs to distribute the noise across all states while in the $\ell_\infty$-norm the computation of perturbation for individual state is independent. Figure 8 illustrates an episode of a Walker agent with and without attack. Our cMBA approach is able to make the agent fall at the 255-th timestep with the episode length and rewards for each setting as: No attack$(661, 2513.57)$, Uniform$(403, 1144.77)$, Gaussian$(455, 1637.16)$, **Ours$(253, 759.85)$**. From the noise values in Figure 7, our cMBA method appears to put noise on a few selective states at each timestep, similar to the Gausian noise setting.

**Additional experiments using the approach in Lin et al. (2020) for continuous action spaces:** In this experiment, we follow the approach in Lin et al. (2020) in the Ant (4x2) environment where we train an adversarial policy for one agent trying to minimize the total team reward while the remaining agents use the trained MARL policy. The adversarial policy is trained for 1 million timesteps. The results using this approach compared with our approach and the two baselines are presented in Figure 11.

From Figure 11, our cMBA approach outperforms Lin et al. (2020)'s approach as it is able to achieve lower team rewards when attacking the same agent. We also note that Lin et al. (2020)'s approach is better than the two baselines using Uniform and Gaussian noise.

**Effect of using $\|\cdot\|_1$ budget constraint:**    In this experiment, we replace the $\|\cdot\|_\infty$ by the $\|\cdot\|_1$ for budget constraint. Using $\|\cdot\|_1$ constraint will produce a significant different adversarial noise pattern

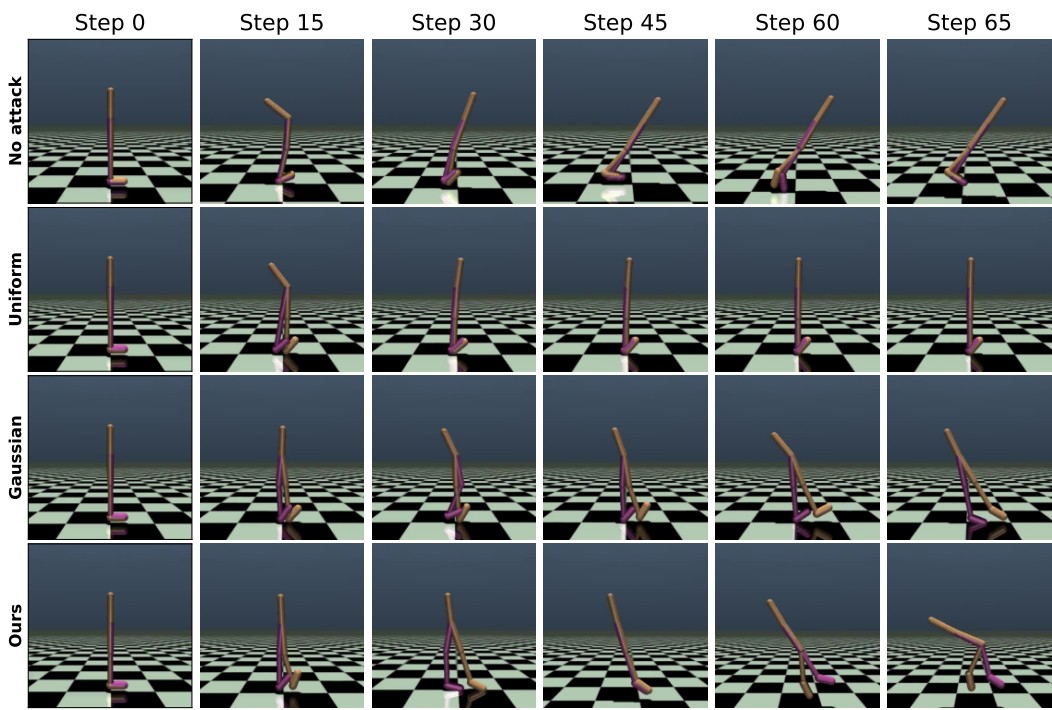

Figure 5: Various attacks on Agent 0 in Walker (2x3) environment with $\|\Delta s\|_\infty \leq 0.2$.

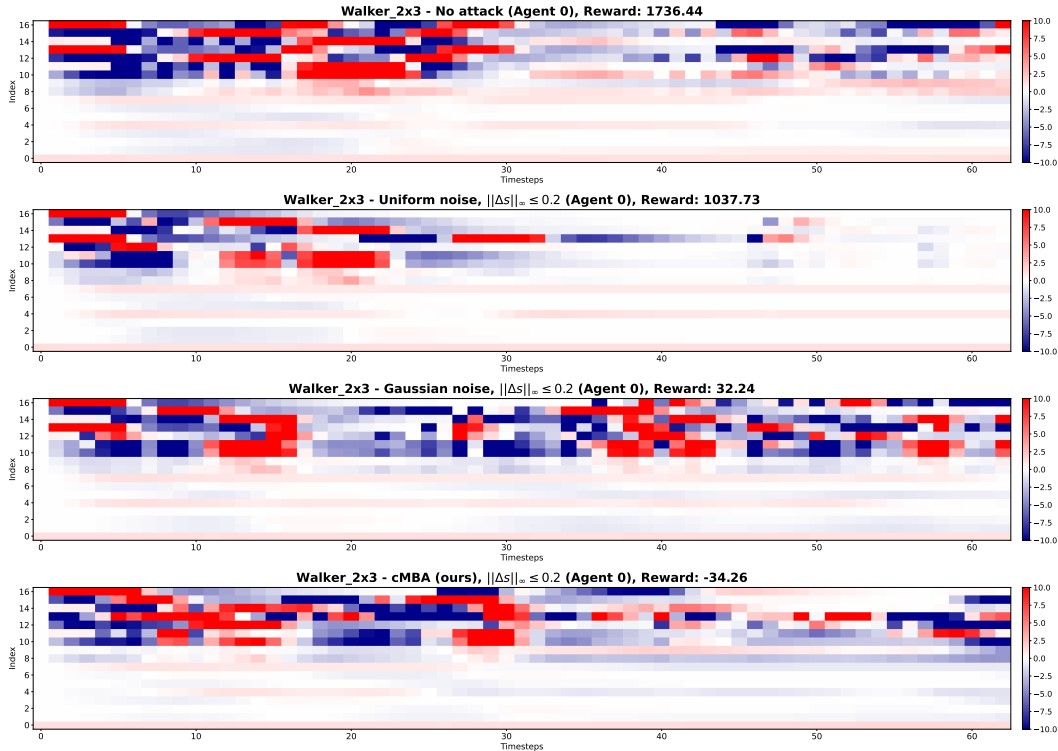

Figure 6: Record of state values in an episode under different attack on Agent 0 in Walker (2x3) environment with $\|\Delta s\|_\infty \leq 0.2$.

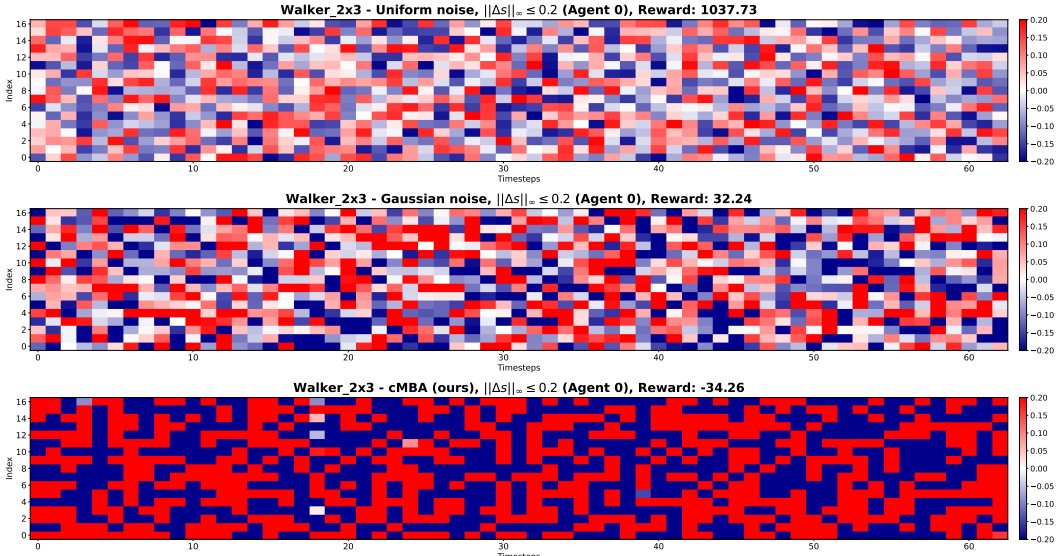

Figure 7: Record of noise values in an episode under different attack on Agent 0 in Walker (2x3) environment with with $\|\Delta s\|_\infty \leq 0.2$.

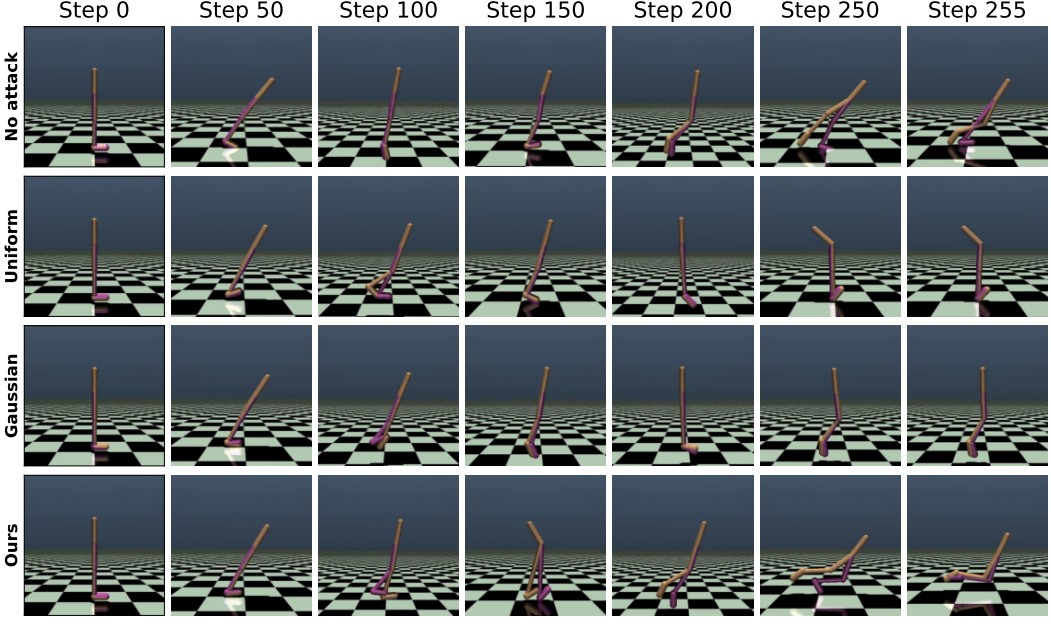

Figure 8: Various attacks on Agent 0 in Walker (2x3) environment with $\|\Delta s\|_1 \leq 0.5$.

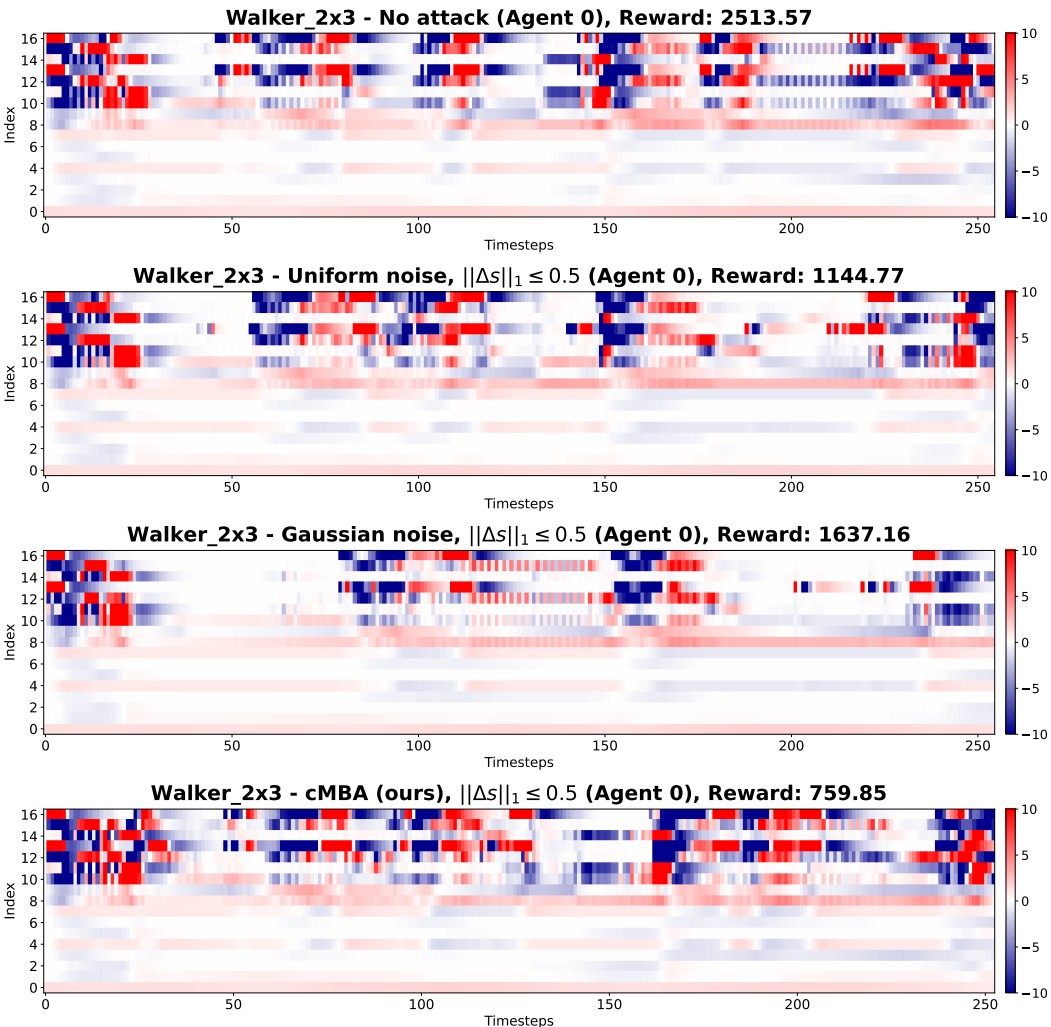

Figure 9: Record of state values in an episode under different attack on Agent 0 in Walker (2x3) environment with with $\|\Delta s\|_1 \leq 0.5$.

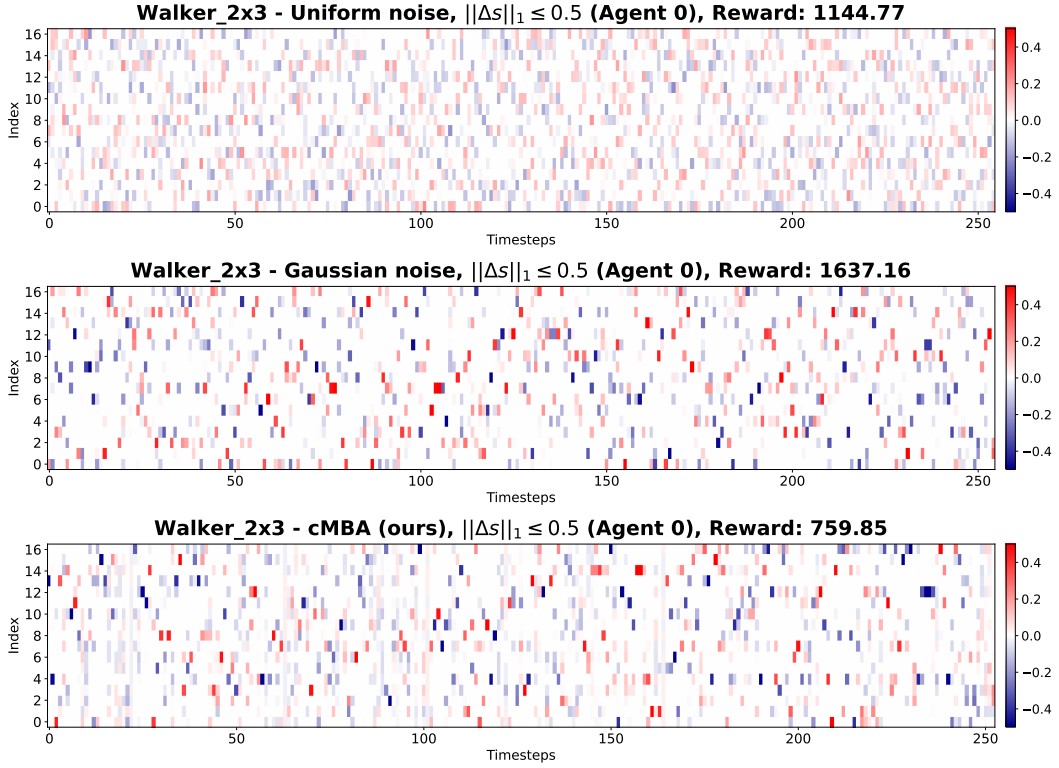

Figure 10: Record of noise values in an episode under different attack on Agent 0 in Walker (2x3) environment with with $\|\Delta s\|_1 \le 0.5$.

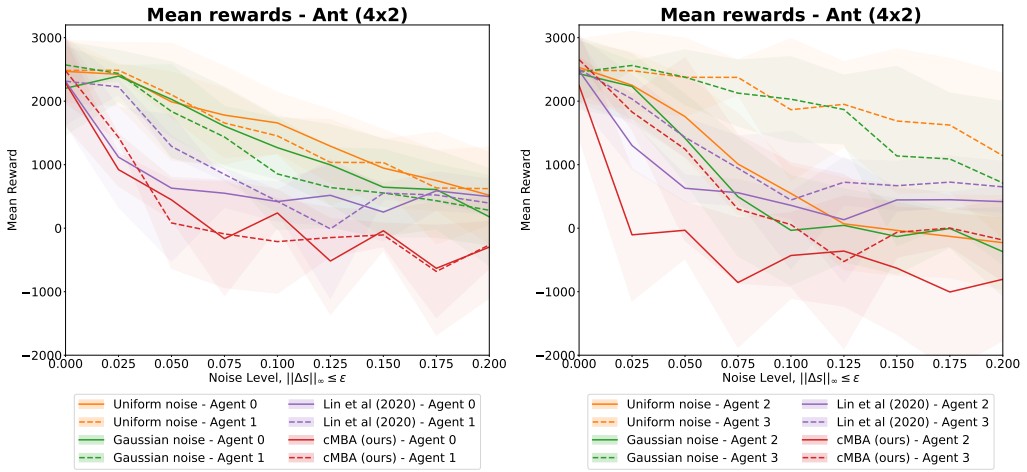

Figure 11: Adversarial attacks using $\|\cdot\|_\infty$ budget constraint in Ant (4x2) environment.

compared to $\|\cdot\|_\infty$ as producing adversarial noise when using $\|\cdot\|_\infty$ is independent for each state while it is not the case for $\|\cdot\|_1$. We run different attacks on two environments, HalfCheetah (6x1) and Ant (4x2), and the results are presented in Figure 12 and Figure 13. From these figures, cMBA outperforms other baselines. In particular, it is able to achieve much lower rewards under smaller budget constraint which shows the advantage of our approach.

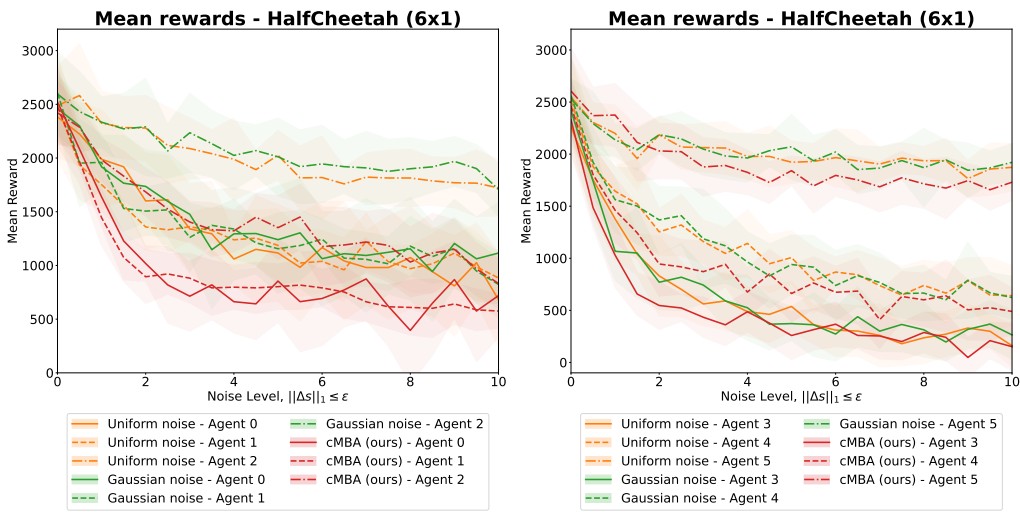

Figure 12: Adversarial attacks using $\|\cdot\|_1$ budget constraint in HalfCheetah (6x1) environment.

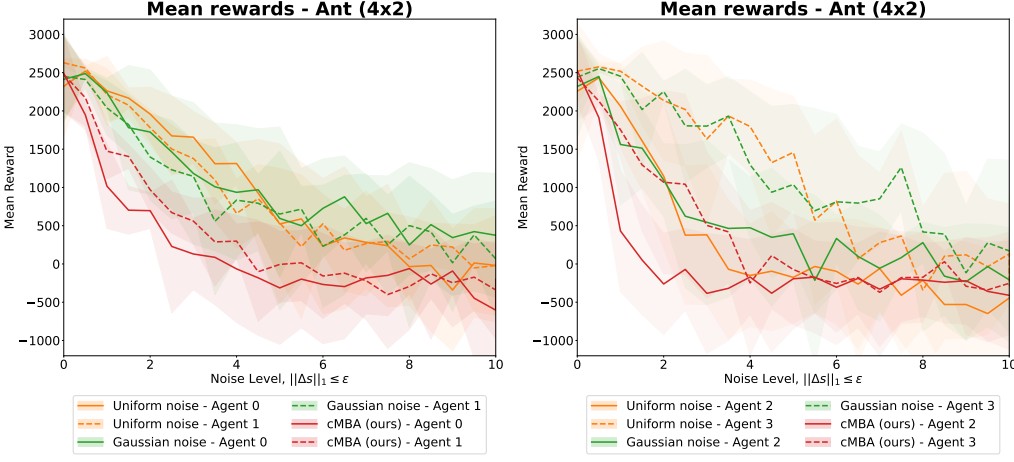

Figure 13: Adversarial attacks using $\|\cdot\|_1$ budget constraint in Ant (4x2) environment.

**Adversarial attacks using dynamics model with various accuracy:** In this test, we compare the performance of our attack when using trained dynamics model with different mean-squared error (MSE). We use the Ant (4x2) environment and consider the following 3 dynamics models:

- The first model is trained using 1 million samples for 100 epochs and select the model with the best performance. We denote this model "1M - Best epoch".

- The second model is trained using 1 million samples for only 1 epoch. We denote this model "1M - 1st epoch".

- We further reduce the number of samples used for training the dynamic model to only 200,000 and train for only 1 epoch. We denote this model "200K - 1st epoch".

These models are evaluated on a predefined test set consisting of 100,000 samples. The test MSE of these models are 0.33, 0.69, 0.79, respectively, with the initial test MSE of 1.241.

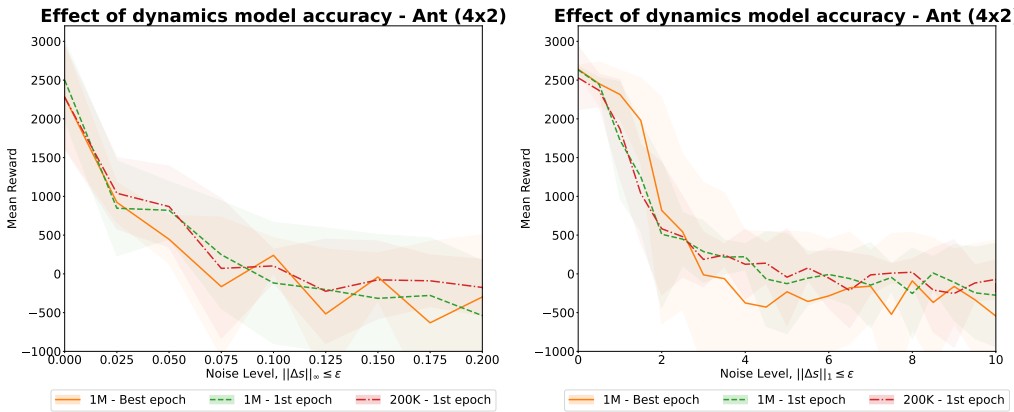

Figure 14: Adversarial attacks using 3 dynamics models in Ant (4x2) environment.

Figure 14 depicts the attacks using these three models on the same agent in Ant (4x2) environment using $\|\cdot\|_\infty$ budget constraint. Interestingly, the dynamics model trained with only 200,000 samples for 1 epoch can achieve comparable performance with the other two models using much more samples.

**Comparison between attacking all agents vs attacking 1 agent:** In this experiment, we compare performance of different approaches when attacking one victim agent or simply perturbing all agents under the same budget constraint. We compare different approaches in the Ant (4x2) environment. In particular, we use the two baselines and our cMBA approach to attack each agent out of 4 agent, denoted as (agent $i$) for $i = 0, 1, 2, 3$. We also use these approach in which we simultaneously perturb the inputs to all agents, denoted as (4 agents). In addition, we also illustrate the performance of our victim selection scheme when we only want to attack either 1 or two agents at the same time. We report the mean and standard deviation of rewards using these approach with several values of budget level $\varepsilon$ in Table 1 where we use $\|\cdot\|_\infty$ for the budget constraint. From Table 1, the performance is getting better as the attacker is stronger, i.e. more agents are attacked at the same time. This makes sense as for $\|\cdot\|_\infty$, the adversarial noise for each agent can be computed independently.

Now, we consider the $\|\cdot\|_1$ budget constraint. This is a more interesting setting as the total noise budget is fixed the adversarial noise of one agent will affect one from the other. The results are shown using several values of budget level $\varepsilon$ in Table 2. We can observe that in this case attacking more agents is not always effective.

Table 1: Comparison between adversarial attacks on a single agent and all agents in Ant (4x2) environment under $\| \cdot \|_\infty \leq \varepsilon$ budget constraint.

| | Rewards: Mean (standard deviation) | | | |
|---|---|---|---|---|
| Methods | $\varepsilon = 0.025$ | $\varepsilon = 0.05$ | $\varepsilon = 0.075$ | $\varepsilon = 0.1$ |
| Uniform noise (agent 0) | 2332 (594) | 2296 (88) | 1773 (941) | 1542 (554) |
| Uniform noise (agent 1) | 2233 922) | 2033 (565) | 1579 (856) | 1293 (385) |
| Uniform noise (agent 2) | 2028 (695) | 1713 (649) | 988 (650) | 322 (652) |
| Uniform noise (agent 3) | 2579 (66) | 2484 (73) | 2159 (623) | 2208 (279) |
| Uniform noise (4 agents) | 2089 (450) | 1093 (753) | 299 (781) | 62 (921) |
| Gaussian noise (agent 0) | 2450 (134) | 2030 (405) | 1660 (663) | 1067 (536) |
| Gaussian noise (agent 1) | 2204 (709) | 1992 (337) | 1447 (406) | 1141 (381) |
| Gaussian noise (agent 2) | 2256 (297) | 1313 (977) | 287 (923) | 154 (570) |
| Gaussian noise (agent 3) | 2550 (83) | 2331 (567) | 1927 (1250) | 2115 (312) |
| Gaussian noise (4 agents) | 2022 (303) | 761 (1083) | 86 (688) | -133 (809) |
| cMBA (agent 0) | 923 (340) | 244 (737) | 516 (737) | -8 (306) |
| cMBA (agent 1) | 1417 (48) | 553 (269) | 108 (303) | -60 (309) |
| cMBA (agent 2) | 227 (955) | -652 (1097) | -241 (660) | -631 (911) |
| cMBA (agent 3) | 2024 (87) | 861 (1298) | 567 (1012) | -172 (725) |
| cMBA (4 agents) | 1116 (379) | 314 (734) | 72 (407) | -239 (281) |
| Leaned Victim Selection (1 agent) | -165 (805) | 624 (103) | 256 (201) | -2 (123) |
| Learned Victim Selection (2 agent) | 405 (308) | -176 (853) | -588 (1026) | -325 (730) |

Table 2: Comparison between adversarial attacks on a single agent and all agents in Ant (4x2) environment under $\| \cdot \|_1 \leq \varepsilon$ budget constraint.

| | Rewards: Mean (standard deviation) | | | |
|---|---|---|---|---|
| Methods | $\epsilon = 1.5$ | $\epsilon = 2.0$ | $\epsilon = 2.5$ | $\epsilon = 3.0$ |
| Uniform noise (agent 0) | 2189 (151) | 2093 (148) | 1615 (524) | 1421 (780) |
| Uniform noise (agent 1) | 2039 (139) | 1715 (282) | 1387 (586) | 1243 (303) |
| Uniform noise (agent 2) | 1542 (826) | 1073 (755) | 581 (920) | 436 (317) |
| Uniform noise (agent 3) | 2468 (75) | 2332 (282) | 2306 (131) | 2136 (319) |
| Uniform noise (4 agents) | 2224 (353) | 2037 (412) | 1734 (541) | 1473 (558) |
| Gaussian noise (agent 0) | 1752 (877) | 1676 (650) | 1431 (774) | 1493 (554) |
| Gaussian noise (agent 1) | 1775 (344) | 1442 (619) | 1257 (487) | 1036 (554) |
| Gaussian noise (agent 2) | 1326 (766) | 1049 (666) | 776 (763) | 516. (842) |
| Gaussian noise (agent 3) | 2103 (559) | 2319 (104) | 2086 (480) | 1896 (603) |
| Gaussian noise (4 agents) | 1614 (776) | 1553 (793) | 1544 (568) | 1470 (476) |
| cMBA (agent 0) | 2068 (145) | 1699 (169) | 1432 (136) | 1282 (136) |
| cMBA (agent 1) | 2005 (355) | 1682 (346) | 1505 (222) | 1189 (585) |
| cMBA (agent 2) | 1554 (781) | 1080 (792) | 433 (1202) | 513 (987) |
| cMBA (agent 3) | 2302 (80) | 2032 (528) | 2099 (92) | 1604 (1069) |
| cMBA (4 agents) | 2212 (225) | 1909 (332) | 1671 (212) | 1504 (201) |
| Leaned Victim Selection (1 agent) | 706 (1118) | 355 (635) | -326 (931) | -139 (608) |
| Learned Victim Selection (2 agent) | 1274 (779) | 720 (939) | 152 (787) | 156 (638) |

