# OpenReview forum: "Evaluating Robustness of Cooperative MARL"
_ICLR.cc/2022/Conference — ICLR 2022 Submitted_

### Official Review · Reviewer_j4AK · 2021-10-31

**Correctness:** 3
**Technical Novelty And Significance:** 2
**Empirical Novelty And Significance:** 2
**Recommendation:** 5
**Confidence:** 4

**Main Review:**

1.	The method proposed in the paper has marginal novelty. It looks like the authors combined different existing method, such as model-based attack, MARL, and PGD to come up with a novel framework for evaluating the robustness of the cooperative MARL. While such a combination has not been explored, the work presented in the paper did not really show sufficient and significant contributions, either theoretical or empirical. In terms of theory, this work is lack of some analysis that can guarantee the optimality or convergence. It is okay to not have such analysis, which means that the effectiveness of the proposed scheme highly depends on the empirical evidence. However, the numerical results shown in the paper to me are not very convincing.
2.	Due to the DECENTRALIZED MARL, typically, method would require communication protocol among various agents. However, in the paper, it looks like the author only summed up all agents and solved the optimization problem to attain the optimal solutions. That way, how to guarantee the optimality?
3.	In the paper, the author leveraged the model-based attack. While a learned model for the environment is helpful, how does the model accuracy affect the performance of the proposed approach? Though using deep net to represent the dynamics may be decent for the state prediction, while the nonlinearity could significantly affect the optimization problems. How to evaluate such effect and empirically solve such a potential issue?
4.	In the numerical results, the authors showed different results for Mujoco environment. However, the baselines to me look simple and in the introduction they have mentioned some existing works, so they should have used one or two in their comparison. Additionally, it is unknown to me whether the proposed method can always outperform those two baselines under different constraint budgets. Since I have only seen 0.2 from the supplementary materials. We can also see that the authors only showed a few agents. What about 20? 30? Even more? Though with more agents, it could be difficult to scale in the Mujoco environment. The authors could use the Gridworld to show that. Moreover, what about topologies? For a decentralized method, it is critical to see the performance with different topologies. While this is missing in the work.
5.	The author mentioned in Algorithm 1 the target state. How to determine the target state in theory and practice?
6.	What is $\mathcal{D}_{random}$ and why did the authors need that to form the buffer?
7.	At the beginning, the authors said that each agent observes its own local state. While in remark 3.1, the authors assumed each agent had access to the other agent’s observation. Why not just assuming at the beginning each agent could observe the global state?

****************************************
After carefully reviewing the authors' responses and comments from other reviewers, I decide to keep the score. I think the current draft is still below the acceptance bar and it requires more work to make it technically solid and sound, particularly in how to determine the target state in a more efficient way, mathematically analysis for the framework, and the quantitative impact of dynamic accuracy on the performance.


**Summary Of The Paper:**

This paper presented a novel method for cooperative MARL to evaluate the robustness under the adversarial states. Specifically, the authors leveraged a model-based approach to perform adversarial attacks on states for MARL with continuous action and state spaces. To represent well the environment, they used a deep neural network to predict the future states. Subsequently they solved an optimization problem with the learned dynamics to produce small perturbations. They also discussed a few strategies to optimally select the victim agents for the attack. To validate the proposed method, the authors implemented numerical experiments on multi-agent Mujoco tasks to show the effectiveness of the proposed method. The investigated topic in this work is quite interesting yet challenging, which remains an active research area. Overall, the paper is easy to follow and well written. However, the authors should pay attention to a few points I have raised in the following to make the paper more technically solid and sound.

**Summary Of The Review:**

Overall, I think the paper still requires substantial work to make sufficient contributions. Particularly, the authors need to address some aforementioned issues, such as marginal novelty and weak numerical results, to make the paper technically solid and sound. Hopefully my comments can help out.

---

> ### Author Response · Authors · 2021-11-23
> **Response to Reviewer j4AK (part 2)**
>
> 4. In the numerical results, the authors showed different results for Mujoco environment. However, the baselines to me look simple and in the introduction they have mentioned some existing works, so they should have used one or two in their comparison. Additionally, it is unknown to me whether the proposed method can always outperform those two baselines under different constraint budgets. Since I have only seen 0.2 from the supplementary materials. We can also see that the authors only showed a few agents. What about 20? 30? Even more? Though with more agents, it could be difficult to scale in the Mujoco environment. The authors could use the Gridworld to show that. Moreover, what about topologies? For a decentralized method, it is critical to see the performance with different topologies. While this is missing in the work.
>
> **Response**: We have added additional results where we use the $\|\cdot\|_1$ budget constraint which produces very different noise pattern compared to the $\|\cdot\|_{\infty}$ constraint. We show that our cMBA approach can achieve lower team reward under smaller budget constraints and outperform other baselines by a large margin.
>
> In our experiments, we consider different environments using up to 6 agents. We agree with the reviewer that scaling with MuJoCo is hard. However, from current results, we observe that the advantage of this approach is getting better as the number of total agents grows as seen in Figure 1.
>
> We are not sure about the use of topologies in adversarial attack. It may be useful when training MARL agents. For adversarial attack, the attacks are performed at each agent independently so we believe topology is not crucial in adversarial attack in MARL.
>
> 5. The author mentioned in Algorithm 1 the target state. How to determine the target state in theory and practice?
>
> **Response**: Currently, we do not have a general procedure to specify this target state. In practice, one needs to have knowledge of the state space of the environment in order to determine a set of global states (e.g. agents’ heights or velocities) that are directly correlated to the total team reward. We need to make sure that assigning these states to specific values will bring down the total team reward. Another approach is to have a separate function approximator for the team reward which takes current state as inputs. Using this function approximator, one can solve an optimization problem to find the appropriate state that minimizes the estimated team reward and use that as the target state.
>
> 6. What is $\mathcal{D}_{random}$ and why did the authors need that to form the buffer?
>
> **Response**: $\mathcal{D}_{random}$ represents a dataset consisting of transitions generated by a random policy, a policy that takes random action at each timestep. The training dataset consists of transitions sampled from a trained policy and a random policy. The motivation of using random policy to sample is to avoid overfitting the dynamics model to the trained policy.
>
> 7. At the beginning, the authors said that each agent observes its own local state. While in remark 3.1, the authors assumed each agent had access to the other agent’s observation. Why not just assuming at the beginning each agent could observe the global state?
>
> **Response**: Remark 3.1 is related to the case where we want to optimally select a set of victim agents so we are assuming that each agent has access to the global state. In the case where the victim agents are preselected, we do not need this assumption and the adversarial attack can be performed at each agent independently of others.

---

> ### Author Response · Authors · 2021-11-23
> **Response to Reviewer j4AK (part 1)**
>
> We thank the reviewer for your helpful comments to improve our paper. Please see our responses for each of the comment below.
>
> 1. The method proposed in the paper has marginal novelty. It looks like the authors combined different existing method, such as model-based attack, MARL, and PGD to come up with a novel framework for evaluating the robustness of the cooperative MARL. While such a combination has not been explored, the work presented in the paper did not really show sufficient and significant contributions, either theoretical or empirical. In terms of theory, this work is lack of some analysis that can guarantee the optimality or convergence. It is okay to not have such analysis, which means that the effectiveness of the proposed scheme highly depends on the empirical evidence. However, the numerical results shown in the paper to me are not very convincing.
>
> **Response**: We note that although model-based attack has been performed in single agent RL, our approach considers the victim selection problem which is unique to the MARL setting. We also emphasize that we aim at formulating the model-based attack to obtain simple problems which can be solved by existing methods.  As stated in the paper, the formulations (1) or (6) can be solved by the projected gradient descent (PGD) method. The method has a convergence rate of $\mathcal{O}\left(\frac{1}{k}\right)$. One can also use an accelerated variant of PGD which achieves $\mathcal{O}\left(\frac{1}{k^2}\right)$ rate. In our experiments, we approximately solve the problem (7) by running PGD for 20 iterations. In addition, apart from the results on $\ell_{\infty}$-norm budget constraint, we have added additional experiments using $\ell_{1}$-norm budget constraint which further illustrates the advantage of our approach. We are able to show that our cMBA approach achieves lower reward under limited budget constraints. Lastly, as suggested by another reviewer, we add another experiment to evaluate our approach under various accuracy when training dynamics models. Furthermore, we add extra experiments to include another baseline as in Lin et al. (2020) for continuous action and experiments using the $\ell_1$-norm budget constraints. The reviewer can refer to Figure 11 to Figure 14 in the Appendix and the highlighted text for more details.
>
> [1] Y. Nesterov. Introductory lectures on convex optimization: A basic course, volume 87 of Applied Optimization. Kluwer Academic Publishers, 2004.
>
> 2. Due to the DECENTRALIZED MARL, typically, the method would require communication protocol among various agents. However, in the paper, it looks like the author only summed up all agents and solved the optimization problem to attain the optimal solutions. That way, how to guarantee the optimality?
>
> **Response**: We want to emphasize that we do not perform training MARL agents in our paper. The goal of formulation (1) and (6) is to produce appropriate adversarial noise to be added to victim agents’ input to lower the total team reward. These formulations are constrained nonconvex problems and can be solved by PGD to obtain a first-order stationary point. If this is not what the reviewer is referring to, it would be great if the reviewer can further clarify this comment.
>
> 3. In the paper, the author leveraged the model-based attack. While a learned model for the environment is helpful, how does the model accuracy affect the performance of the proposed approach? Though using deep net to represent the dynamics may be decent for the state prediction, while the nonlinearity could significantly affect the optimization problems. How to evaluate such effect and empirically solve such a potential issue?
>
> **Response**: This is a good point. As also suggest from Reviewer **55ss**, we have added an experiment where we use 3 dynamics models trained at different MSE level and perform the attack using both $\ell_{\infty}$-norm and $\ell_1$-norm budget constraint. You can refer to the end of Appendix B in our revised version.

---

### Official Review · Reviewer_jFMh · 2021-11-02

**Correctness:** 2
**Technical Novelty And Significance:** 2
**Empirical Novelty And Significance:** 2
**Recommendation:** 3
**Confidence:** 4

**Details Of Ethics Concerns:**

The reviewer does not see obvious ethics concerns.

**Main Review:**

***Soundness***

The main concern of the reviewer is about the novelty and applicability of the proposed method.

1) The proposed attacking method depends on a pre-trained dynamics model. Training such a model may be costly in multi-agent settings.

2) The most frustrating point about the proposed method is that the target state needs to be predefined, which largely limits the applicability of the proposed method.

3) For selecting the victim agents, the proposed method needs local information of all the agents. This assumption is less realistic in multi-agent settings. Moreover, together with 2), this assumption simplifies the difficulty of attacking MARL algorithms.

4) The presented results are not surprising because the instability of MARL has been well studied. As reported in the paper, random noise and randomly selected victim agents can also reduce performance significantly.The reviewer thinks that perturbing local observations is not a promising research direction. The reviewer holds that a simple attacking method where the gradients of observation with respect to value function is used to perturb the observations can have significant influence.

5) The proposed method has access to the policy of all agents. Such a white-box attacking method is less applicable.

***Evaluation***

The baselines are not powerful. Please show the results of the baseline described in Soundness (4).

***Clarity***

This paper is not well written, and there are various grammatical errors throughout the paper. For example, in section 3.2, above equation 1, “$A$ is the set of concatenated states.” I guess this should be actions?

**Summary Of The Paper:**

The paper proposes a method to attach cooperative MARL strategies on tasks with continuous action spaces. A dynamics model is trained, based on which agents’ local observations are perturbed so that agents select actions that lead to the least-rewarding states.

**Summary Of The Review:**

The problem under research is interesting, but the proposed method can access parameters of all agents’ policies, can modify observations of all agents, and depends on pre-defined target states. These assumption hurts the novelty and applicability of the paper.

---

> ### Author Response · Authors · 2021-11-23
> **Response to Reviewer jFMh**
>
> We thank the reviewer for your helpful comments to improve our paper. Please see our responses below for each of your comment.
>
> 1. The proposed attacking method depends on a pre-trained dynamics model. Training such a model may be costly in multi-agent settings.
>
> **Response**: Regarding training dynamics model. We first collect a desired number of transitions then use supervised learning to train the dynamic model. Let us provide more detail on the cost of training a dynamic model. For the Ant (4x2) environment, the data collection takes about 1 hour for 1 million timesteps and training takes about 30 minutes to have a dynamic model trained for 100 epochs. Meanwhile, we use Lin et al. (2020)’s approach to retrain the environment with one victim agent while the other agents use the pre-trained policy. This approach takes 5-6 hours to train for 1 million timesteps. We also measure the time to perform the two attacks at each transition under the same budget constraint. For Lin et al. (2020), the time it takes to perform adversarial attacks for 10 episodes is 173 seconds while ours take 386 seconds. Although the running time of our approach is longer than Lin et al. (2020), it is still negligible and our approach does achieve better performance as in Figure 11.
>
> 2. The most frustrating point about the proposed method is that the target state needs to be predefined, which largely limits the applicability of the proposed method.
>
> **Response**: We agree with the reviewer that the current target state needs to be defined beforehand. For actor-critic policies, you can use the critic which estimates the Q-function for team rewards and find a target state that minimizes this reward function and use it in model-based attack. However, our approach also works when you do not have such information. To make our approach more general, we tradeoff the fact that we do require the knowledge of the state spaces so that we can select a desired target state that is directly related to low team reward.
>
> 3. For selecting the victim agents, the proposed method needs local information of all the agents. This assumption is less realistic in multi-agent settings. Moreover, together with 2), this assumption simplifies the difficulty of attacking MARL algorithms.
>
> **Response**: This assumption results from observation from real-life application. Imaging a team of people moving a large box, each person needs to know how other people are doing which means the person requires the knowledge of other people’s states. Analogously, we are considering the cooperative MARL (c-MARL) setting where a team of agents cooperatively complete a task. Therefore, we believe the assumption that each agent has access to others’ local states is still valid.
>
> 4. The presented results are not surprising because the instability of MARL has been well studied. As reported in the paper, random noise and randomly selected victim agents can also reduce performance significantly.The reviewer thinks that perturbing local observations is not a promising research direction. The reviewer holds that a simple attacking method where the gradients of observation with respect to value function is used to perturb the observations can have significant influence.
>
> **Response**: We note that the random noise performs better with higher budget constraints, e.g. for $\epsilon \ge 0.2$ in Figure 1. The state spaces of these environments range from [-1,1] and a budget constraint of 0.2 is rather large. Our cMBA approach is able to achieve lower reward within smaller budget constraints. This behavior can be further observed in Figure 12 and 13 which we have added to show the performance of cMBA using $\ell_1$-norm constraint. Overall, we believe that our approach is more beneficial under limited budget constraints as compared to random noise approach.
>
> 5. The proposed method has access to the policy of all agents. Such a white-box attacking method is less applicable.
>
> **Response**: Similar to our response to Reviewer **55ss**, we want to note that we do not really need to have access to the model’s parameter, we just need to have a mechanism to access the gradient of the output action with respect to the state input. In particular, one can use a black-box MARL policy to generate proper action given the current state. Then we can use techniques to estimate the gradient including forward difference or Gaussian kernel (Maggiar et al. 2018) and still be able to solve the sub-optimization problem to find the suitable adversarial noise.

---

### Official Review · Reviewer_55ss · 2021-11-02

**Correctness:** 3
**Technical Novelty And Significance:** 2
**Empirical Novelty And Significance:** 2
**Recommendation:** 5
**Confidence:** 3

**Main Review:**

The following are some of my concerns and questions:
1. The main issue is related to the novelty of this work. Previous works (Gleave et al. 2020, Lin et al. 2020) have looked at attacking MARL. Thus it is no surprise that MARLs are vulnerable to perturbation of the agents’ observation. In fact, as shown in the evaluation section, random noise seems to have comparable (or sometimes better) performance compared to the proposed method, which further emphasizes the instability of the cooperative MARL algorithms. In fact, the brittleness of cooperative strategies is richly documented in game theory literature, which is not addressed in this paper.

2. From a practical point of view, this work also has a few shortcomings. The attack method proposed in this work is a white-box attack, which requires the attacker to have access to the model’s parameter, which limits the applicability of the attack.

3. The proposed method in fact does not directly minimize the team reward, but the users need to manually define a target state as part of the attack objective. Defining the target states can be challenging for some use cases, and hence it is in fact non-trivial to evaluate the robustness of the MARL algorithm.

4. Gradient-based attack methods such as FGSM and JSMA are not compared in this work. Although the authors mentioned that the target action is not available in the continuous action space, one can actually back prop from the centralized critic toward to actor policy with an objective to minimize the team reward. How do those methods compare to your proposed method?

5. One of the contributions in this work is the victim selection during the attack. However, it seems the attacker needs to have observations of all agents and be able to modify the observation of any agent. This can be challenging during the execution phase of MARL where the agents are physically distributed. Can you provide some concrete applications of this attack? What are the assumptions of the attack?

6. This attack requires training a model-based state prediction model. How much data is needed to train such a model? How much time is required for training? Does it assume the attacker has access to the underlying transition function?


**Summary Of The Paper:**

This work focuses on attacking cooperative multi-agent reinforcement learning using a model-based method. First, a model is trained to predict the next state based on the current state and action. Then PGD is used to find a small perturbation on the current state so that the next state is closed to a target state. The target state is selected using heuristic methods and they are used as a surrogate objective for minimizing the team reward. A victim agent selection method is also proposed for selecting the best target action.


**Summary Of The Review:**

Although this work is looking at an important problem related to the robustness issue of MARL, its technical contribution fail short in term of the theoretical analysis, novelty, and evaluations.

---

> ### Author Response · Authors · 2021-11-23
> **Reponse to Reviewer 55ss (part 2)**
>
> 4. Gradient-based attack methods such as FGSM and JSMA are not compared in this work. Although the authors mentioned that the target action is not available in the continuous action space, one can actually back prop from the centralized critic toward to actor policy with an objective to minimize the team reward. How do those methods compare to your proposed method?
>
> **Response**: If we do backprop through the centralized critic to find the action that minimizes the team reward, we do require access to other agent’s information. In our approach, we only need to require access to other agent’s information when we solve the subproblem to find the optimal set victime agents. If the set of victim agents are prespecified, we solve the subproblem for each agent in a decentralized fashion to generate the suitable adversarial noise.
>
> Following your suggestion, we have added additional experiments to include an equivalent approach of Lin et al. (2020) for continuous action spaces where we train an adversarial policy and use this to generate a desired action for the victim agent. Then we use iterative FGSM to find the proper adversarial noise added to the agent input given the desired action. The results are presented in Figure 11.
>
> 5. One of the contributions in this work is the victim selection during the attack. However, it seems the attacker needs to have observations of all agents and be able to modify the observation of any agent. This can be challenging during the execution phase of MARL where the agents are physically distributed. Can you provide some concrete applications of this attack? What are the assumptions of the attack?
>
> **Response**: We first note that we are considering a cooperative MARL (c-MARL) setting where multiple agents are teaming to complete a task. Consider when two people are moving a desk from point A to point B. One person needs to observe the current state of the other person so that the coordination is on point. And suppose we use robots instead to perform this task, we also need to ensure that one robot has knowledge of another’s state for better coordination. Therefore, we believe this assumption is valid in the c-MARL setting.
>
> 6. This attack requires training a model-based state prediction model. How much data is needed to train such a model? How much time is required for training? Does it assume the attacker has access to the underlying transition function?
>
> **Response**: This is a great question. We have added an experiment where we use 3 dynamics models trained at different MSE level and perform the attack using both $\ell_{\infty}$-norm and $\ell_1$-norm budget constraint. You can refer to the highlighted text near the end of Appendix B in our revised version.

---

> ### Author Response · Authors · 2021-11-23
> **Reponse to Reviewer 55ss (part 1)**
>
> We thank the reviewer for your helpful comments to improve our paper. Please see our responses to each of your comment below.
>
> 1. The main issue is related to the novelty of this work. Previous works (Gleave et al. 2020, Lin et al. 2020) have looked at attacking MARL. Thus it is no surprise that MARLs are vulnerable to perturbation of the agents’ observation. In fact, as shown in the evaluation section, random noise seems to have comparable (or sometimes better) performance compared to the proposed method, which further emphasizes the instability of the cooperative MARL algorithms. In fact, the brittleness of cooperative strategies is richly documented in game theory literature, which is not addressed in this paper.
>
> **Response**: We want to note that although Gleave et al. (2020) considers multi agent setting, they do not consider **cooperative** multi agent reinforcement learning (c-MARL) where multi agents team together to maximize a total team reward function. As indicated in our title “evaluating robustness of cooperative MARL”, the focus of this work is on the vulnerability of **cooperative** MARL, and to the best of our knowledge, Lin et al. (2020) is the only work considering c-MARL setting as ours.
>
> However, our approach is fundamentally different from one in Lin et al. (2020). We do not need to solve a RL (policy optimization) problem to find an adversarial policy which could be computationally expensive (4 times longer than training the dynamics model in the Ant (4x2) environment). In fact, we only need to solve a supervised learning problem to train the dynamics model. We note that both approaches require interacting with the environment. Our work contributes to the setting of adversarial attack on MARL by showing that the adversarial noise can also be computed using a learned dynamics model.
>
> We note that the random noise performs better with higher budget constraints, e.g. for $\varepsilon \geq 0.2$ in Figure 1. The state spaces of these environments range from [-1,1] and a budget constraint of 0.2 is rather large. Our cMBA approach is able to achieve lower reward within smaller budget constraints. This behavior can be further observed in Figure 11 and 12 which we have added to show the performance of cMBA using $\ell_1$-norm constraint. Overall, we believe that our approach is more beneficial under limited budget constraints as compared to random noise approach.
>
> We are not sure about the literature of cooperative strategies in game theory that the reviewer mentioned. We would appreciate it if the reviewer can provide some reference and we will provide more discussion on this.
>
> 2. From a practical point of view, this work also has a few shortcomings. The attack method proposed in this work is a white-box attack, which requires the attacker to have access to the model’s parameter, which limits the applicability of the attack.
>
> **Response**: We want to note that we do not really need to have access to the model’s parameter, we just need to have a mechanism to access the gradient of the output action with respect to the state input. Therefore, our approach is applicable for *black-box attacks* too. In particular, one can use a black-box MARL policy to generate proper action given the current state. Then we use certain techniques to estimate the gradient including forward difference or Gaussian kernel (Maggiar et al. 2018) and still be able to solve the sub-optimization problem to find the suitable adversarial noise.
>
> 3. The proposed method in fact does not directly minimize the team reward, but the users need to manually define a target state as part of the attack objective. Defining the target states can be challenging for some use cases, and hence it is in fact non-trivial to evaluate the robustness of the MARL algorithm.
>
> **Response**: Thank you for your comment. We agree that the current framework needs to have a certain understanding of the state space of the environment. However, we emphasize that each agent in a c-MARL environment will have access to a set of global states (robot’s height or velocity) and these states are directly correlated to the total team reward. Therefore, the global state values corresponding to low team reward will be used as target state in our approach without the need of training an adversarial policy to generate this target state.

---

### Official Review · Reviewer_1ANY · 2021-11-02

**Correctness:** 3
**Technical Novelty And Significance:** 2
**Empirical Novelty And Significance:** 2
**Recommendation:** 3
**Confidence:** 3

**Main Review:**

I like the idea of using a learned dynamics model to reduce the need for environment interactions when computing proposed attacks.

I'm not so sure about the constraint that you need to prespecify a target state that leads to low performance. In some sense, if you have to pre-specify this, you have to predeterimine the type of attack you want your adversary to make, as opposed to letting it learn what are the best types of attacks given the domain. The paper against which you compare, Lin et al. (2020), seems more general to me in that they try to minimize the reward of the team. This is a broader approach in that your reward function could simply be I(s ==s_target) if you wanted your adversary to reach a certain state as in your work.

"Note that we cannot directly apply methods such as the fast gradient sign method (FGSM) or JSMA (an attack using saliency map) as in Lin et al. (2020), since those methods require a “target” action which is not available in continuous action space." — why is the case? If I understand correctly, the target action in Lin et al. (2020) is proposed by a RL agent policy, which could easily be a continuous action policy no? To that end, it's unclear to me why their method wouldn't work in a continuous action space if you substitute policy gradient for q-learning. That being said, it could be interesting to compare to their method with and without a learned model to replace the need for interactions with the environment.

Why is victim selection important? What the real world instances where you'd have an adversary be able to select a different agent at each timestep but not be able to simple perturb all of them?

This method would be more interesting to me if it allowed for and experimented with discrete or mixed continuous-discrete action spaces.

Small comments:

Overall I found the grammar and writing to be somewhat poor and distracting.

To me, using language "dynamic model" is more unclear than "dynamics model". The latter I think is more commonly used.

"neural network of target policy" → "neural network of **the** target policy"

"another line of research tackle" → "another line of research tackle**s**"

"problem on DRL with continuous action space**s**"

"where A is the set of concatenated states" — I think this should be "concatenated actions".

**Summary Of The Paper:**

Proposes an adversarial attack on continuous cooperative multi-agent settings. Given a target low performing state, the adversary chooses an observation perturbation by trying to minimize the distance from the next state (as predicted by a dynamics model) to the target state via projected gradient descent. They also experiment with optimizing for and choosing a limited number of agents to perform the adversarial action at each timestep. There experiments are in multi-limbed mujoco tasks such as Walker, HalfCheetah, and Ant, and show that their method causes the agent to receive lower reward than a random adversary.

**Summary Of The Review:**

Overall I felt this paper was below the acceptance threshold primarily because I felt the experiment domains rather limited and disappointing that the only baseline was random noise (and that the method seems to do only marginally better than random noise). I also felt that prior methods could work in continuous action spaces if very slightly modified, so I felt this paper could be much stronger if it experimented with and compared to those variants.

---

> ### Author Response · Authors · 2021-11-23
> **Response to reviewer 1ANY (part 2)**
>
> 4. Why is victim selection important? What the real world instances where you'd have an adversary be able to select a different agent at each timestep but not be able to simple perturb all of them?
>
> **Response**:
> In the original manuscript, we consider attacking only 1 agent -- it is because Lin et al. (2020) also uses the same setting. One of the major contribution in this paper is that, we demonstrate that when the attack budget is fixed (e.g. the attacker can only attack 1 agent), with *proper* selection on the vulnerable agent, we could make the modern c-MARL system perform worse, as demonstrated in Figure 4 that 3 out of 4 cases are successful. However, we note that our method could be applied to attacking multiple agents (>1) and the same time by defining the victim set properly in Eqn 1, which should make the c-MARL agents perform even worse (as now the adversary is *stronger* and *more powerful*). We have an additional experiment where we compare different approaches when attacking all agents or 1 agent under the same budget constraint. This is demonstrated in our experiments presented in Table 1.
>
> Regarding your question, if an attacker can perturb all the agents’ observations, then this is the strongest adversary and we expect the resulting system performance would be the worst (as the attacker is very powerful); while if an attacker can only perturb a subset of agents’ perturbation (e.g. in Lin et al. (2020) and our original manuscript, only 1 agent can be attacked), then it’s a weaker attacker and the harm to the system should be smaller. In the real-world scenario, we think a possible setting is when the ML system is equipped with some protection mechanism that does not allow multiple (or some fixed threshold) agents to be perturbed, then the attacker’s power will be limited.
>
> To sum up, the purpose of the setting in this paper is to evaluate how *vulnerable* is the c-MARL algorithm (i.e. even under the “*weaker*” attacker setting, where *only 1 agent can be attacked*). We would like to know would the modern c-MARL agents still be vulnerable to the “*weak*” adversary”? And unfortunately the answer is Yes, as demonstrated in this paper and Lin etal 2020.
>
> 5. This method would be more interesting to me if it allowed for and experimented with discrete or mixed continuous-discrete action spaces.
>
> **Response**: Thank you for your suggestion. The focus of this work is to explore the model based attack using continuous action space. We’ll include your suggestion on a more challenging setting in our future work.
>
> 6. Small comments: Overall I found the grammar and writing to be somewhat poor and distracting.
> **Response**: We have fixed these typos in our revision.

---

> ### Author Response · Authors · 2021-11-23
> **Response to reviewer 1ANY (part 1)**
>
> We thank the reviewer for their helpful comments to improve our paper. Please see our reponses to your comment below.
>
> 1. I like the idea of using a learned dynamics model to reduce the need for environment interactions when computing proposed attacks.
>
> **Response**: Thank you for your comment.
>
> 2. I'm not so sure about the constraint that you need to prespecify a target state that leads to low performance. In some sense, if you have to pre-specify this, you have to predeterimine the type of attack you want your adversary to make, as opposed to letting it learn what are the best types of attacks given the domain. The paper against which you compare, Lin et al. (2020), seems more general to me in that they try to minimize the reward of the team. This is a broader approach in that your reward function could simply be I(s ==s_target) if you wanted your adversary to reach a certain state as in your work.
>
> **Response**: In Lin et al. (2020), they need to retrain a policy for the victim agent to minimize the total team reward while remaining agents generate actions using a pre-trained policy. From this trained adversarial policy, the agent can specify a target state as input to an adversarial method. Different from their approach, we first train a dynamics model and use it to find the adversarial noise added to the agent's input. Although our approach is still a two-step method similar to Lin et al. (2020), we do not need to solve an RL problem to find an adversarial policy as we only solve a simple supervised learning problem with the sampled transitions. Note that the setting in Lin et al. (2020) requires complete access to an agent to obtain an adversarial policy while our approach does not. This means Lin et al. (2020) needs to gain control of the agent to retrain the adversarial policy before using it to generate the desired target state. For our approach, to train the dynamics model, we can collect the transitions as the agents are performing their task normally then use cMBA to generate the adversarial noise. Our approach is similar to the man-in-the-middle attack in cyber security. In addition, our approach is also more applicable in settings when the dynamics model is already provided.
>
> 3. "Note that we cannot directly apply methods such as the fast gradient sign method (FGSM) or JSMA (an attack using saliency map) as in Lin et al. (2020), since those methods require a “target” action which is not available in continuous action space." — why is the case? If I understand correctly, the target action in Lin et al. (2020) is proposed by a RL agent policy, which could easily be a continuous action policy no? To that end, it's unclear to me why their method wouldn't work in a continuous action space if you substitute policy gradient for q-learning. That being said, it could be interesting to compare to their method with and without a learned model to replace the need for interactions with the environment.
>
> **Response**: We want to note that in order to apply FGSM, one needs to have a target action. Lin et al. (2020) proposes to train an adversarial policy to generate this target action. We want to emphasize that our approach and Lin et al. (2020) are two-steps method whereas our first step is only to solve a supervised learning problem while Lin et al. (2020) solves a policy optimization problem which takes longer to train. The reviewer can refer to response #1 to Reviewer **jFMh** below for an example of training time that we conducted. Moreover, we have added Figure 11 to include Lin et al. (2020)’s approach for the Ant (4x2) environment and show that our approach achieves the best performance our of the four approaches. As an example, at budget level $\epsilon = 0.1$, the corresponding reward for all approaches are: 239.73 (cMBA - ours), 419.16 (Lin et al., 2020), 1267.96 (Gaussian noise), and 1657.49 (Uniform noise). Last but not least, we note that Lin et al. (2020)’s approach also requires interacting with the environment to train an adversarial policy, and incur higher cost than our proposed attack because training an adversarial policy is costly (e.g. 6 hours for 1 M samples) compared to our approach learning dynamics model (1.5 hrs).

---

### Decision · Program_Chairs · 2022-01-20

**Decision:**

Reject

**Comment:**

Although the problem studied in the paper is interesting, all the reviewers believe that the current draft has limited technical contributions. Moreover, there are serious issues with the writing and presentation of the work. Also the experiments are rather limited and their results are not significant. I strongly recommend the authors to take the reviewers' comments into account and improve different aspects of their work for future conferences.